# Sensorimotor Cortical Activity during Respiratory Arousals in Obstructive Sleep Apnea

**DOI:** 10.3390/ijms24010047

**Published:** 2022-12-20

**Authors:** Katharina Bahr-Hamm, Nabin Koirala, Marsha Hanif, Haralampos Gouveris, Muthuraman Muthuraman

**Affiliations:** 1Sleep Medicine Center, Department of Otolaryngology, University Medical Center of Johannes Gutenberg University Mainz, 55131 Mainz, Germany; 2Haskins Laboratories, Yale University, New Haven, CT 06511, USA; 3Movement Disorders and Neurostimulation, Biomedical Statistics and Multimodal Signal Processing Unit, Department of Neurology, University Medical Center of Johannes Gutenberg University Mainz, 55131 Mainz, Germany; 4Neural Engineering with Signal Analytics and Artificial Intelligence (NESA-AI), Department of Neurology, University Hospital Würzburg, 97080 Würzburg, Germany

**Keywords:** respiratory arousal, obstructive sleep apnea, cortical activity, spectral entropy, approximate entropy

## Abstract

Intensity of respiratory cortical arousals (RCA) is a pathophysiologic trait in obstructive sleep apnea (OSA) patients. We investigated the brain oscillatory features related to respiratory arousals in moderate and severe OSA. Raw electroencephalography (EEG) data recorded during polysomnography (PSG) of 102 OSA patients (32 females, mean age 51.6 ± 12 years) were retrospectively analyzed. Among all patients, 47 had moderate (respiratory distress index, RDI = 15–30/h) and 55 had severe (RDI > 30/h) OSA. Twenty RCA per sleep stage in each patient were randomly selected and a total of 10131 RCAs were analyzed. EEG signals obtained during, five seconds before and after the occurrence of each arousal were analyzed. The entropy (approximate (ApEn) and spectral (SpEn)) during each sleep stage (N1, N2 and REM) and area under the curve (AUC) of the EEG signal during the RCA was computed. Severe OSA compared to moderate OSA patients showed a significant decrease (*p* < 0.0001) in the AUC of the EEG signal during the RCA. Similarly, a significant decrease in spectral entropy, both before and after the RCA was observed, was observed in severe OSA patients when compared to moderate OSA patients. Contrarily, the approximate entropy showed an inverse pattern. The highest increase in approximate entropy was found in sleep stage N1. In conclusion, the dynamic range of sensorimotor cortical activity during respiratory arousals is sleep-stage specific, dependent on the frequency of respiratory events and uncoupled from autonomic activation. These findings could be useful for differential diagnosis of severe OSA from moderate OSA.

## 1. Introduction

Cortical arousals are strongly associated with sleep-disordered breathing [1]. For a long time, it has been assumed that after reaching a certain arousal threshold, arousals and respiratory events are linked and terminates a potentially life-threatening apnea or hypopnea [2]. However, this assumption has been challenged by recent findings observing the temporal correlation between respiratory arousals and reopening of the upper airway. These studies have shown that in 10–40% of obstructive events, no arousal occurs during or around the opening of the airway [3,4]. Moreover, in about 20% of the patients, arousals were detected only after reopening the airway [4]. Furthermore, the major dilator muscle of the respiratory tract, the genioglossus muscle, was observed to be active even before the onset of an arousal [5], the threshold of which varied individually [6,7]. The duration of respiratory events also seems to have an impact on cardiovascular endpoints in patients with obstructive sleep apnea (OSA) [8,9]. Patients with short respiratory events appear to have a lower arousal threshold and may be predisposed to increased ventilatory instability. This instability was shown to increase the likelihood of adverse health outcomes and even predict the mortality rate [10]. The premature interruption of a respiratory event by an arousal might further lead to a higher arousal index, which makes it more difficult to enter deeper sleep stages resulting in daytime tiredness [11,12] and increased risk of obstructive events during sleep [13]. Hence, the role that arousals play in the microstructure of sleep, specifically in sleep-disordered breathing, appears to be essential, however, not fully understood.

The intensity of the respiratory cortical arousals (RCA) captured using electroencephalography (EEG) has been previously demonstrated as a pathophysiological trait in obstructive sleep apnea (OSA) patients [14,15]. Although clinical experts are able to categorize arousal based on changes in EEG amplitude and frequency spectrum, they are unable to capture EEG microstructure features. Hence, quantitative computer-aided dynamic metrics used to analyze these microstructural features of human respiratory cortical arousals may help in automating and assisting the clinical experts. One such metric could be the EEG signal entropy, which is the amount of uncertainty in the signal pattern and is roughly equivalent to the amount of information contained in that signal [16]. This metric has been shown in previous studies to be useful for sleep-related and other neurological disorders. Approximate entropy has been shown to detect and characterize seizures in patients with epilepsy [17], classify neurodegenerative disorders from healthy controls [18], detect sleep hypopnea [19] and evaluate sleep quality in obstructive sleep apnea-hypopnea syndrome [20]. Recently, approximate entropy was also used in a nonlinear model combined with polysomnography (PSG) features like nasal airflow and thoracic movements to predict the presence and severity of OSA [21]. Similarly, spectral entropy has been used to monitor the evolution of seizure and epileptiform activity in epilepsy patients [22,23], observe abnormal neural activity in Alzheimer’s disease and MCI patients [24], and quantify the impact and regulation of anesthesia [25,26]. In this study, we investigated the entropies related to respiratory arousals in OSA to examine the central motor activity in the brain during sleep-associated respiratory events. We have previously shown the involvement of cortical sensorimotor areas in motor control pathways associated with obstructive respiratory events in OSA patients using EEG recordings of standard polysomnography (PSG) [27]. Here, we hypothesize that arousal EEG signals have a specific pattern, which would be repeated in the different time interval around the arousals. Hence, we investigated if for time intervals—during as well as five seconds before and after the respiratory arousals. A particular frequency distribution was repeated with high probability and was specific to OSA severity or sleep stages.

## 2. Results

### 2.1. EEG-Based Markers

EEG analysis showed a significant difference (*p* < 0.0001) in the area under the curve (AUC) during the respiratory cortical arousals (RCA) between the two groups (Figure 1). The AUC for the severe OSA group (mean: 0.399 ± 0.052) was significantly decreased compared to that of the moderate OSA group (mean: 0.599 ± 0.050) indicating that the two OSA severity groups may be distinguished by means of their arousal microstructure.

The approximate entropy showed a significant increase (*p* < 0.0001) during (mean *Group A (moderate OSA)*: 0.696 ± 0.069, *Group B (severe OSA)*: 0.395 ± 0.055) and after (mean *Group A*: 0.59 ± 0.11, *Group B*: 0.44 ± 0.08) the arousal compared to before (mean *Group A*: 0.40 ± 0.06, *Group B*: 0.29 ± 0.05) the arousal for both OSA groups (Figure 2). Importantly, the amount of entropy was significantly reduced in severe OSA patients compared to moderate OSA patients for all (pre, during and post) arousal phases. In contrast, the spectral entropy showed a significant (*p* < 0.05) decrease during (mean *Group A*: 0.38 ± 0.039, *Group B*: 0.39 ± 0.025) and after (mean *Group A*: 0.29 ± 0.26, *Group B*: 0.40 ± 0.05) the arousal compared to before (mean *Group A*: 0.41 ± 0.06, *Group B*: 0.44 ± 0.09) the arousal for both OSA groups. Moreover, spectral entropy was significantly higher for severe OSA patients compared to the moderate group. This indicates that the variation in the arousal structure contributes to the differentiation of the moderate and severe OSA even only when considering the EEG data and without considering respiratory events. Moreover, this prominent difference appears to not only involve the arousal itself, but also to extend to the time windows immediately preceding and following the arousal.

For the analysis performed on individual sleep stages, the approximate entropy was significantly higher for the severe OSA patients in each of the sleep stages (mean during N1: *Group A*: 0.40 ± 0.056, *Group B*: 0.59 ± 0.060; N2: *Group A*: 0.21 ± 0.060, *Group B*: 0.29 ± 0.039; REM: *Group A*: 0.34 ± 0.030, *Group B:* 0.47± 0.040) compared to moderate OSA patients (Figure 3a). In addition, for both groups, the entropy was highest during N1 sleep, followed by REM and N2 sleep stages (Figure 3a) most probably because of the larger external influences during these sleep stages compared to N2.

### 2.2. Association to Clinical Measures

We observed significant negative correlations (r = −0.27, *p* = 0.023) between the apnea–hypopnea index (AHI) and SpEN; pre approximate entropy and ESS (r = −0.29, *p* = 0.013); and post sample entropy and BMI (r = −0.28, *p* = 0.017) for moderate OSA patients during the respiratory arousals. In addition, significant positive correlations between sample entropy (pre) and sex (r = 0.25, *p* = 0.035), and between approximate entropy (during) and BMI (r = 0.29, *p* = 0.011) were found. Furthermore, the AUC and the ODI were negatively correlated (r = −0.25, *p* = 0.038), as well as the arousal-EEG sample entropy for the N2 sleep stage with ESS (r = −0.29, *p* = 0.010). There was no significant correlation between ApEN and AHI in any sleep stage or during arousals (Figure 4).

In contrast to the moderate OSA patient cohort, the severe OSA cohort showed overall lower correlations between most of the analyzed parameters. We found only two significant correlations: one negative between AHI and AUC (r = −0.38, *p* = 0.002) and a positive correlation between sleep stage N1 sample entropy and BMI (r = 0.33, *p* = 0.008). These associations clearly indicate the efficacy of these EEG-based markers in clinical settings.

### 2.3. Scientific Control

In order to investigate whether the respiratory events were associated with autonomous nervous or cortical mechanisms, we performed further analyses. More specifically, we computed the approximate entropy values using heart rate variability (HRV) to determine whether there was an influence from the autonomous system in the computed measures. We found no significant difference between ApEN computed using HRV between any of the sleep stages (Figure 3b). This finding, combined with the results obtained using EEG entropy, clearly strongly supports the hypothesis that the autonomic nervous system has no effect on the analyzed arousals.

## 3. Discussion

In this study, we computed the area under the curve of the respiratory arousals, as well as the spectral and approximate entropy using the EEG data in the temporal course of RCA, as a quantitative marker to classify OSA severity. We evidenced significant difference in both the entropies between severe and moderate OSA, computed during and five seconds before and after the RCA. Based on these measurements obtained during, before and after respiratory arousals, we were able to distinguish accurately between moderate and severe OSA patient groups. Additionally, and most importantly, we were able to quantify the sleep stage-specific variability of entropy and could provide strong evidence for a cortical, rather than autonomic, driver of respiratory arousals by further testing the HRV-associated approximate entropy. 

EEG data obtained from polysomnography recordings are routinely used to detect relevant OSA subtypes and further provide more details on sleep stages and clinical consequences of OSA such as sleepiness, chronic cognitive changes, metabolic changes and cardiovascular consequences [28,29,30]. Previous studies have used a range of EEG analyses including spectral analysis [31,32], wavelet transformation [33], cyclic alternating pattern analysis [34], K-complex detection [35,36], sleep spindles analysis [36] and several other EEG-derived signals and features relevant to OSA [14,37]. However, these studies often focused on either a specific sleep stage or a complete sleep period. They do not consider arousal as an important feature of obstructive sleep apnea, which is generally not the focus of the study. A previous study has also used an entropy measure as the EEG index to determine the sleep quality in obstructive sleep apnea–hypopnea syndrome patients using full PSG recordings [20]. However, in this study we analyzed only the RCA EEG signal, which has an advantage of not only revealing a major pathology of obstructive sleep apnea but also does not require the data recorded for the entire night. Moreover, we used both approximate and spectral entropy in this study to complement the advantages of both methods. SpEn is based on the frequency spectrum of a signal and has the particular advantage that the entropies of several frequency sections can be precisely separated from each other [38]. However, ApEn quantifies the predictability of subsequent amplitude values of a signal and has the advantage of enabling the complexity or regularity of systems to be captured quantitatively [39].

Here, we found that severe OSA patients showed a significantly lower AUC of arousal microstructure compared to moderate OSA patients. The lower AUC suggests a more monotonous sequence of arousals and possibly indicates a habituation effect of the cortex because of repeated RCA that are most frequent in severe OSA. We hypothesize this based on findings from previous studies which have indicated that habitual response would reduce the dimensional complexity of the EEG signal, for example, in working memory load tasks [40].

We observed that approximate entropy was higher after and during the arousal compared to before the arousal. In contrast, the spectral entropy was lower. A previous study that utilized spectral entropy measurements showed that it increased with increasing sleep quality and efficiency in patients with sleep apnea [20]. This is contrary to our results, which showed an increase in spectral entropy in severe OSA stages compared to mild and moderate ones. An explanation for this was not found at first. However, it should be noted that Ohisa et al. did not further classify the different OSA severities and focused on sleep efficiency rather than disease severity. On the other hand, similar to our findings, Gutierrez-Tobal et al. showed in their study that patients with obstructive apnea showed a higher EEG frequency and SpEn measures [41]. Even though they attribute this to thalamic inhibition in obstructive sleep apnea, the reason why the thalamus function is inhibited remains unclear.

Sleep stage analysis showed a reduction in ApEn from stage N1 to N2 in both mild and severe OSA. This is congruent with previous findings and the assumption that with deeper sleep the complexity of brain activity decreases [42]. REM sleep, on the other hand, although characterized by a higher threshold of awakening ability, is known to have brain activity similar to stage N1 or wakefulness, and therefore has higher entropy in both groups. Furthermore, it may be suggestive of a robust arousal signal pattern generator in N2 stage sleep and less robust generators in REM and even less in N1. This fact may imply that arousal generation during REM and (even more) in N1 may be subjected to external influences to a much larger extent than during N2. Additionally, this instability is much more prominent in severe compared to in moderate OSA, proving the sleep disruption caused by respiratory (apneic or hypopneic) events, that by definition happen more often in severe OSA (higher AHI). 

The correlation analysis showed a few correlations between AHI and AUC/ApEn/SpEn. However, the correlations were sparse, and parameters differed between the group A and B. Thus, we cannot conclude that one of the three metrics could depict the AHI in some way.

Previous studies have shown success in differentiating and predicting the severity of the OSA patients using signals other than EEG recorded during PSG. Kaimakamis et al. developed and tested a predictive model for the presence and severity of OSA using a simple linear equation, utilizing nonlinear features extracted from respiratory recordings; nasal cannula flow and thoracic belt movement [21]. Similarly, Liu et al. used the full night blood oxygen values from PSG recordings to compute approximate entropy to differentiate obstructive sleep apnea–hypopnea syndrome patients from normal controls (snorers in this case) and show the association to the apnea–hypopnea index (AHI) [43]. Moreover, they evidenced that it was possible to identify different degrees of severity of OSA on the basis of the degree of nocturnal oxygen deficiency.

The entropy measurements for heart rate variability (HRV) as applied in our study have also been applied before. Liang et al. used entropy, in this case the sample entropy, to evaluate the HRV and complexity of the autonomic nervous system in OSA patients. The changes were significantly different between the normal and OSA groups. In this case, it was shown once again that entropy was able to identify and quantify OSA. In contrast to our study, the entropy was neither applied to EEG measurements nor to respiratory arousals [44]. 

Sleep stage analysis with entropy was also applied by Sharma et al. in 2022. According to the authors, the entropy model is robust and accurate to identify sleep stages in both healthy individuals and patients suffering from sleep disorders. In contrast to our study, Sharma et al. included healthy individuals and thus a control. On the other hand, the algorithm needed information from various channels, which were EEG, electrooculogram (EOG) and electromyogram (EMG). Apart from sleep stage analysis, which was performed with accuracy, there was no intention to classify sleep disorders such as OSA [45]. 

In general, an advantage of ApEn can be seen in the fact that it proves to be stable against artifacts of different sizes by applying the filter “r” [46]. Valuable information can be obtained from the rather high quantity of data points. Again, the advantage of our study is the focus on respiratory events, which means that, in contrast to the previously mentioned studies, no recording of the entire night is required.

In this study, we found significant and highly relevant results for clinical application which could be adopted by sleep and neurological laboratories or even for at-home sleep apnea testing that includes EEG. 

A limitation of our study lies in the fact that it is a retrospective data analysis, which furthermore lacks a control group, since the data of patients presenting with OSA were analyzed. It would be interesting to look at the entropy behavior of healthy individuals and also patients with mild OSA (AHI = 5–15/h), provided they show at least a small amount of respiratory arousals during the course of a polysomnography night. Nonetheless, healthy participants exhibit much less frequent respiratory arousals; hence, a direct comparison with patients with moderate and severe OSA might not have been possible at all. As a result, our findings do not allow any conclusions for OSA patients with an AHI greater than 5 and less than 15 per hour. Nevertheless, it is remarkable that, using this method, the two similar groups of patients could be distinguished with such accuracy just using the EEG signals from two electrodes. 

A major focus of science and industry today is in looking for an easier, cheaper and patient-friendlier method as a reliable substitute for polysomnography. The described method of EEG entropy still requires EEG measurements during the night. However, the perspective that can be gained from this study is that fragments alone, such as respiratory arousals or the differences in entropy before and after an arousal, allow the diagnosis and quantification of OSA. Thus, the completeness of an entire polysomnography (over a period of at least 6 h, as well as with all its conduction channels) might no longer be needed with further development of this technology in the future. Realistically, this method can only currently be seen as a surrogate marker to strengthen the diagnosis and support the existing metrics to identify the severity of respiratory events. Especially if PSG quality is bad/not adequate (in terms of recording quality and duration), this method could still lead to an adequate diagnosis and estimation of OSA severity.

## 4. Materials and Methods

### 4.1. Data Acquisition

Raw EEG data for 102 OSA patients (32 females, mean age 51.6 ± 12 years) were obtained from C3 and C4 electrodes during polysomnography (PSG) according to American Academy of Sleep Medicine (AASM) 2007 guidelines [47]. All data were obtained retrospectively from patients’ data collected during the standard clinical PSG procedures in our tertiary university hospital department. The ethical approval was obtained from the local Institutional Review Board (Approval Nr. 2018-13942) to use this data for research purposes, and data acquisition was conducted according to the principles of the Declaration of Helsinki [48]. Exclusion criteria included the intake of prescription sleep medication or psychotropic drugs, patients who had had a stroke, periodic leg movement disorder or restless legs syndrome, NYHA stage III or IV heart failure and patients with stage III or IV chronic obstructive pulmonary disease (COPD) [49]. Moreover, patients with neurophysiological diseases, such as multiple sclerosis, Parkinson’s disease, epilepsy and narcolepsy, were also excluded from the analysis [50]. Data were retrieved from only those patients who spent two nights in the sleep laboratory as only the raw data from the second night is used for clinical routine, as well as for the analyses here. This procedure is necessary to avoid distortions in the results due to the so-called first-night effect [51].

For all patients, data from electroencephalography (EEG), electro-oculography (EOG), electromyography (EMG) and electrocardiography (ECG), along with flow (nasal respiratory flow), thorax and abdominal movements (respiratory effort), oxygen saturation, snoring sounds, the body position using a position sensor and video recordings using an infrared camera to detect and differentiate sleep-related movement disorders and epilepsies [47] were recorded. The EEG data were recorded using silver/silver chloride electrodes placed at C3, C4, F4 and O2 locations based on the EEG 10–20 system [52]. The sampling rate was 200 Hz, with low-pass filters of 0.3 Hz and high-pass filters of 35 Hz applied. All PSG data were collected using the Alice^®^ LE system (Philips Respironics, as supplied by Loewenstein Medical, Bad Ems, Germany) and was subsequently converted and stored in the European Data Format (EDF). The EEG data obtained from C3 and C4 electrodes were used for analysis [27]. The EOG data which recorded vertical and horizontal movements for the right and left eye were collected using self-adhesive electrodes, and ECG collected using 12 disposable electrodes was used to record and monitor the heart activity. A sleep medical expert carried out the sleep stage analysis and scoring of arousals and respiratory events. Additional clinical diagnostic data were extracted, which included patient demographics, body mass index (BMI), Epworth sleepiness scale (ESS) score and primary diagnosis details containing apnea–hypopnea index (AHI), oxygen desaturation index (ODI), total sleep time (TST) and minimal and median oxygen saturation. 

### 4.2. Data Analysis

Out of 102 patients selected for the study, 47 had moderate (respiratory distress index, RDI = 15–30/h) and 55 had severe (RDI > 30/h) obstructive sleep apnea. Patients with moderate OSA were assigned to “*Group A*” and those with severe OSA to “*Group B*”. The mean BMI of the entire patient population was 31.3 kg/m^2^ ± 5.3 kg/m^2^ and thus lies within the range of grade I obesity according to WHO [53]. A total of 99 patients filled in the ESS questionnaire and the mean ESS score was 9.9 ± 4.9. All demographics and clinical parameters are detailed in Table 1. Comparisons between the groups were made by the Kruskal–Wallis test and the unpaired *t*-test. The Kruskal–Wallis test was performed to identify inequalities within the demographic data and thus to detect potential confounders like age or BMI. Twenty RCA were randomly selected per sleep stage in each patient and a total of 10131 RCAs were used for analysis. The data acquisition and methodological pipeline implemented in the study is illustrated in Figure 5.

Sleep epochs were selected only from the complete duration of PSG recordings in order to consider all sleep phases equally. Care was taken to ensure that the PSG epochs were free of artifacts (e.g., body movements) and did not overlap. The EEG signals during, as well as five seconds before and after, each arousal were manually scored and extracted. The EEG signals were pre-whitened (amplitude equalized at all frequencies) and normalized before estimating the pooled spectra [54]. The pooled EEG trace (C3 and C4) was smoothed with a fourth order Butterworth filter, from which the area under the curve was estimated. The raw EEG signals of the respiratory arousals were statistically analyzed using a receiver operating curve (ROC) [55]. Subsequently, the area under the curve (AUC) of the ROC was calculated to determine if the two groups of patients significantly differentiate in the morphology of their respiratory arousals according to severity. In total, we were able to obtain 3199 epochs with RCA for group A and 6932 epochs with RCA for group B for the analyses. Furthermore, the EEG signals were used to compute approximate and spectral entropy. *Approximate entropy* (ApEn) is a metric that depicts the irregularity of the signal and is computed by embedding the signal into the phase space and estimating the rate of increment in the number of phase space patterns within a predefined value, *r*, when the embedding dimension of phase space increases from *m* to *m + 1.* Please refer to studies [38,39] for elaborated mathematical details. *Spectral entropy (SpEn)* quantifies the probability density function (PDF) of the signal power spectrum in the frequency domain and is computed using state entropy (SE) and response entropy (RE) values. SE measures EEG activity up to a frequency of 32 Hz and RE includes frequencies up to 47 Hz for EEG. Please refer to studies [56,57] for mathematical details. In this study, we only used response entropy which was estimated over the frequency range 0.8 to 47 Hz as our spectral entropy parameter.

Group level statistical comparison for both entropy measures was performed using the Kruskal–Wallis test in IBM SPSS (Version 23). The covariates were age, body mass index (BMI), apnea–hypopnea index (AHI), respiratory distress index (RDI), apnea index, hypopnea index, total sleep time (TST), sleep efficiency, arousal index, number of all physiological arousals and number of all respiratory arousals. Moreover, to test if the respiratory arousals were not driven by the autonomic nervous system but rather by cortical activity, we compared approximate entropy values computed using heart rate variability (HRV, as a marker of autonomic nervous system activation) and EEG signals during arousals in all sleep stages. We further investigated a sleep stage (N1, N2, REM)-specific analysis of RCAs using the approximate entropy (ApEn). The N3 sleep stage was excluded as sleep fragmentation and respiratory arousals are less prominent at this stage and OSA is most noticeable in NREM 1, NREM 2 and REM stages [58]. Finally, the Pearson correlation (corrected using Bonferroni correction) was used for identifying significant associations between the estimated entropy variables (ApEN, SpEN and arousal AUC) and the clinical variables (AHI).

## 5. Conclusions

We provide evidence for a significant difference in the area under the curve of the arousal signal and entropy (both spectral and approximate) during respiratory cortical arousals between patients with moderate and severe OSA. The distinction between the two OSA severity grades can be made solely by examining the EEG signal of the cortical respiratory arousals. The entropy increase observed in severe OSA patients suggests strong alterations in sensorimotor cortical activity before, after and during the arousals. Moreover, we provide evidence that the dynamic range of sensorimotor cortical activity during respiratory arousals is sleep stage-specific, dependent on the frequency of respiratory events and—most importantly—uncoupled from autonomic nervous system activation.

These novel metrics might be used as surrogate markers to identify and quantify OSA apart from established PSG parameters. They may also substantially support decisions on OSA diagnosis and OSA severity in everyday practice cases in which the recording quality of the PSG is too poor for diagnosis according to the established standard metrics, which require a full night PSG. Since this study did not include a control group (without any OSA at all), these quantitative EEG markers should be further validated in larger clinical studies with larger cohorts including a broader range of disease severity including OSA and non-OSA individuals.

## Figures and Tables

**Figure 1 ijms-24-00047-f001:**
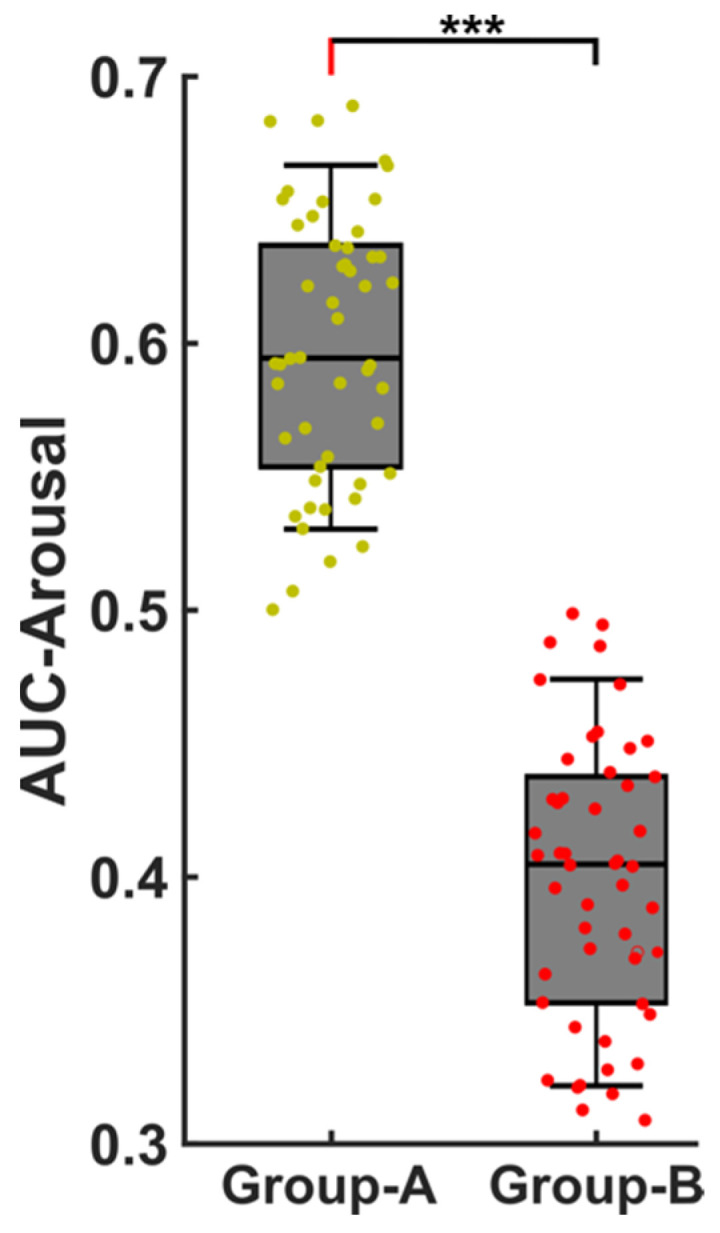
Area under the curve (AUC) for respiratory arousals. Group A: moderate OSA patients, Group B: Severe OSA patients. *** represents significance of *p* < 0.001.

**Figure 2 ijms-24-00047-f002:**
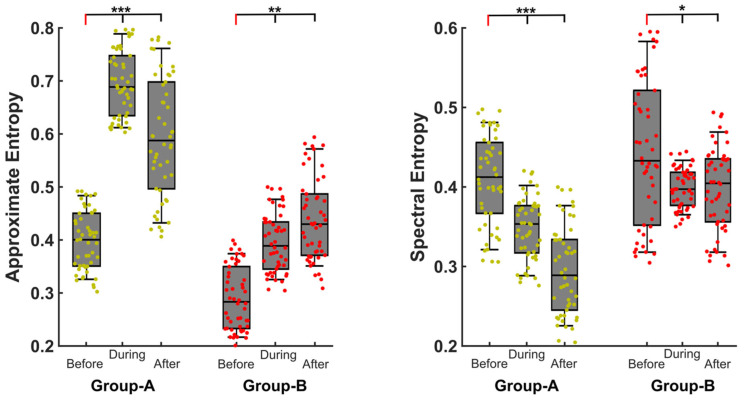
Results of approximate entropy and spectral entropy. Here, Group A: moderate OSA patients, Group B: severe OSA patients. During, before and after refers to the period with respiratory cortical arousals (RCA), and five seconds before and after the arousal, respectively. *** represents significance of *p* < 0.001, ** represents *p* < 0.01, * represents *p* < 0.05.

**Figure 3 ijms-24-00047-f003:**
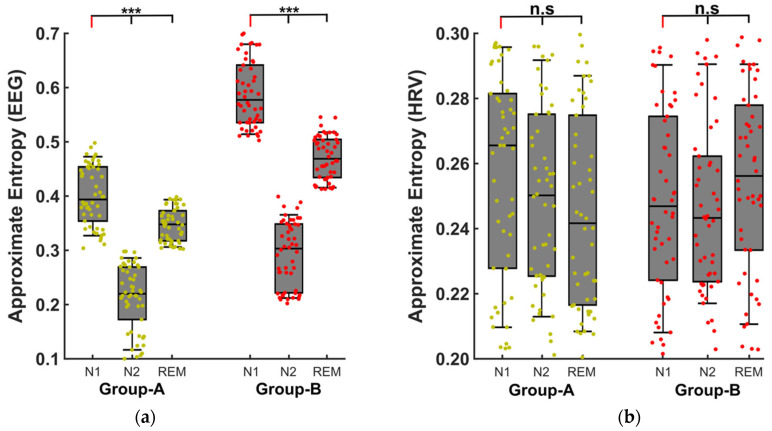
Sleep stage analysis using the approximate entropy of the EEG signal (**a**) and of the heart-rate-variability (HRV) (**b**) during the respiratory arousals in the different sleep stages. Here, Group A: moderate OSA patients, Group B: severe OSA patients, REM: rapid eye movement sleep stage, N1 and N2: non-rapid eye movement sleep stages. *** represents significance of *p* < 0.001, n.s. = non-significant.

**Figure 4 ijms-24-00047-f004:**
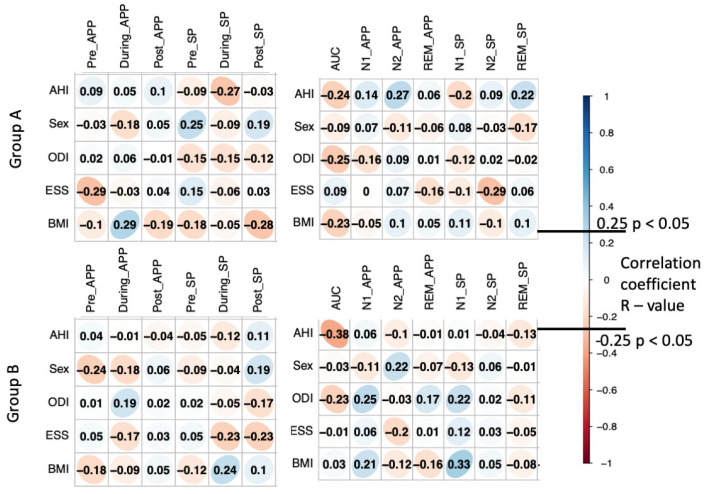
Correlation analysis of Group A and B in the form of R values. AHI = apnea–hypopnea index, ODI = oxygen desaturation index, ESS = Epworth sleepiness score, BMI = body mass index, Pre_APP = approximate entropy before the arousal, During_APP = approximate entropy during the arousal, Post_APP = approximate entropy after the arousal, Pre_SP = spectral entropy before the arousal, During_SP = spectral entropy during the arousal, Post_SP = spectral entropy after the arousal, AUC = area under the curve, N1/N2/REM_APP = approximate entropy in sleep stage N1/N2/REM, N1/N2/REM_SP = spectral entropy in sleep stage N1/N2/REM. The R values of ≤−0.25 and ≥0.25 correspond to a *p*-value < 0.05.

**Figure 5 ijms-24-00047-f005:**
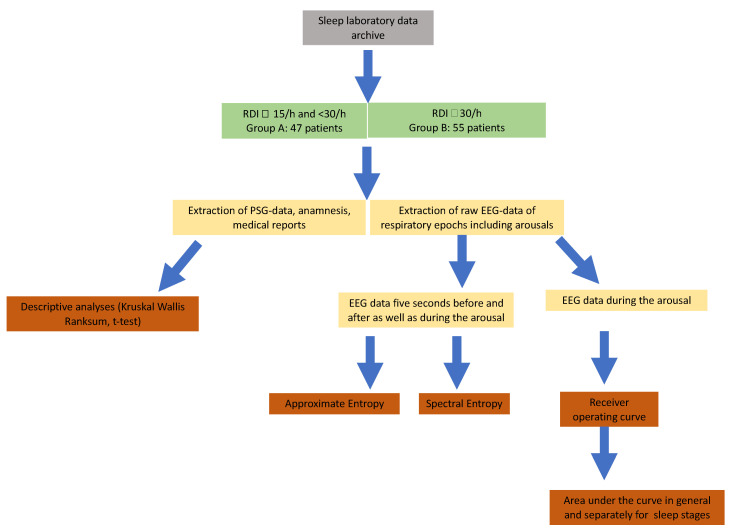
Flowchart of applied methodology.

**Table 1 ijms-24-00047-t001:** Demographic and clinical details of the participants in the study. TST: total sleep time, ESS: Epworth sleepiness scale, BMI: body mass index, REM: rapid-eye-movement sleep stage, NREM = non-rapid-eye-movement sleep stage, ODS: oxygen desaturation index in REM sleep stage, ODS-NREM: oxygen saturation in NREM sleep stages, ArI: arousal index, T90 = sleep time spent with a blood oxygen saturation below 90% in percent. * Clinically significant differences between *Group A* and *Group B*.

	All Participants	Group A (Moderate OSA)	Group B (Severe OSA)	*p* Value
**N (F/M)**	102 (32/70)	47 (20/27)	55 (12/43)	
**Age range (mean)** *years*	27–86 (51.6 ± 12.0)	28–70 (50.17 ± 10.8)	27–86 (52.9 ± 12.9)	0.247
**BMI range (mean)** *kg/m^2^*	19.6–46.6 (31.3 ± 5.3)	20.9–46.5 (30.9 ± 5.6)	19.5–43.2 (31.7 ± 5.1)	0.289
**ESS score range (mean)** *points*	2–21 (9.9 ± 4.9)	3–19 (9.8 ± 4.7)	2–21 (10.2 ± 5.2)	0.398
**TST range (mean)** *minutes*	167–491 (348.5 ± 61.5)	203–49 (362.3 ± 54.2)	167–458 (336.9 ± 65.4)	0.068
**Sleep efficiency range (mean)** *%*	47.9–98.2 (83.4 ± 11.1)	62.6–98.2 (85.7 ± 8.6)	47.9–98.2 (81.5 ± 12.7)	0.145
**ODS range (mean)** *per hour*	1.4–97.5 (27.7 ± 23.0)	1.4–29.8 (12.8 ± 7.3)	8.6–97.5 (40.4 ± 24.2)	<0.001 *
**T90 range (mean)** *%*	0–68.5 (6.6 ± 11.9)	0–27.6 (2.7 ± 5.1)	0–68.5 (10.0 ± 14.8)	0.001 *
**ODS-REM range (mean)** *%*	77–98 (93 ± 3.4)	88–98 (94.2 ± 2.2)	77–96 (92 ± 4.0)	0.001 *
**ODS-NREM range (mean)** *%*	86–97 (93.5 ± 1.9)	91–97 (94.2 ± 1.7)	86–97 (93.0 ± 2.0)	0.004 *
**ArI range (mean)** *per hour*	4.3–93.4 (30.8 ± 16.5)	4.3–42.5 (22.1 ± 7.4)	9.9–93.4 (38.2 ± 18.5)	<0.001 *
**RCA range (mean)**	7–395 (99 ± 72.4)	13–151 (67.3 ± 29.5)	7–395 (126 ± 86.2)	<0.001 *

## Data Availability

The data presented in this study are available on request from the corresponding author. The data are not publicly available due to in-house guidelines.

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
