# Peer review of "Sensorimotor Cortical Activity during Respiratory Arousals in Obstructive Sleep Apnea"

_ijms, 2022, doi:10.3390/ijms24010047_

Round 1
Reviewer 1 Report
Dear authors,
Thank you for studying this potentially clinically useful subject
I have some clinical remarks in order to improve the quality of your work
Look forward to reviewing the revised version
Best regards
Reviewer
Could the authors clarify the clinical significance of this study to the patients please? The majority of OSA patients require a proper overnight polysomnography in order to diagnose OSA on them. Are there specific advantages to the patients where the performance of a polysomnography study would not be necessary then?
I think the lack of control group for a descent comparison of data is a great limitation of this study to set up the standards which indeed is highlighted in the limitation. This should also be expressed in the conclusion section.
Author Response
Could the authors clarify the clinical significance of this study to the patients please? The majority of OSA patients require a proper overnight polysomnography in order to diagnose OSA on them. Are there specific advantages to the patients where the performance of a polysomnography study would not be necessary then?
--
We thank the reviewer for this remark, since this is an issue we discussed among authors extensively. Ideally, we are looking for an easier, cheaper and patient-friendlier method as a reliable substitute for polysomnography. Still, our method of EEG entropy requires EEG measurements during the night. However, the perspective that can be gained from this study is that fragments alone, such as respiratory arousals or the differences in entropy before and after an arousal, allow the diagnosis of OSA, and thus the completeness of an entire polysomnography (over a period of at least 6 hours, as well as with all its conduction channels) might no longer be needed with further development of this technology. Realistically, right now this method can be seen as a surrogate marker to strengthen the diagnosis and support the existing matrix to identify the severity of respiratory events. Especially if PSG-quality is bad/not adequate (in terms of recording quality and duration), this method could still lead to an adequate diagnosis and estimation of severity.
We added our thoughts and this statement to the discussion part, kindly see lines 374-385 and 397-401.
I think the lack of control group for a descent comparison of data is a great limitation of this study to set up the standards which indeed is highlighted in the limitation. This should also be expressed in the conclusion section.
--
The reviewer is right. The analysis was performed retrospectively on OSA patients and the lack of control is definitely a weakness worth mentioning. We tried to point it out more clearly in the discussion part, kindly see lines 359-363 and also mentioned it -as requested- in the conclusion part (401-404).
Reviewer 2 Report
This manuscript is interesting and well written. However, I have some remarks :
1. The area under (a ROC) curve is a measure of the accuracy of a quantitative diagnostic test. A ROC curve is a plot of the true positive rate (Sensitivity) in function of the false positive rate (100-Specificity) for different cut-off points of a parameter. Legend figure 2 described AUC but this is a box-plot.
2. References do not present recently published papers related to this study.
3. The lack of a control group reduced quality of the presented study. A retrospective cohort study refers to existing medical records, the researcher mainly focused on data analysis. The study was not planned.
Author Response
- The area under (a ROC) curve is a measure of the accuracy of a quantitative diagnostic test. A ROC curve is a plot of the true positive rate (Sensitivity) in function of the false positive rate (100-Specificity) for different cut-off points of a parameter. Legend figure 2 described AUC but this is a box-plot.
--
The reviewer is absolutely right with the area under the curve. Actually, the ROC area under the curve is estimated in this study for each individual patient. In order to achieve clarity and to present the data in the same way for all figures, which shows each individual point is a patient in all figures, we have chosen to show it as bar plots. In addition, this way we could see the standard deviation between the two groups.
- References do not present recently published papers related to this study.
--
We thank the reviewer for this remark, although we tried to include recently published studies on this topic, this manuscript has been in progress for quite a while and it is true that the latest literature was not considered. We included two more studies from 2021 and 2022, one dealing with EEG and EMG data for sleep stage analysis and one dealing with entropy and HRV, in the discussion part, kindly see new lines 336-349.
3. The lack of a control group reduced quality of the presented study. A retrospective cohort study refers to existing medical records, the researcher mainly focused on data analysis. The study was not planned.
--
The reviewer is right. The data was retrospectively analyzed and lacks a control group. This is a weakness of the study, which we pointed out more clearly in the discussion and conclusion part. Kindly see line 339-363 and 401-404.